# Dietary Intakes Are Associated with HDL-Cholesterol in Survivors of Childhood Acute Lymphoblastic Leukaemia

**DOI:** 10.3390/nu11122977

**Published:** 2019-12-05

**Authors:** Sophia Morel, Devendra Amre, Emma Teasdale, Maxime Caru, Caroline Laverdière, Maja Krajinovic, Daniel Sinnett, Daniel Curnier, Emile Levy, Valérie Marcil

**Affiliations:** 1Research Centre, Sainte-Justine University Health Center, Departments of Nutrition, Université de Montréal, Montreal, QC H3T 1C5, Canada; sophia.morel@umontreal.ca (S.M.); emma.teasdale@umontreal.ca (E.T.); emile.levy@recherche-ste-justine.qc.ca (E.L.); 2Institute of Nutrition and Functional Foods, Laval University, Quebec City, QC G1V 0A6, Canada; 3Research Centre, Sainte-Justine University Health Center, Departments of Pediatrics, Université de Montréal, Montreal, QC H3T 1C5, Canada; devendra.amre@umontreal.ca (D.A.); caroline.laverdiere@umontreal.ca (C.L.); maja.krajinovic@umontreal.ca (M.K.); daniel.sinnett@umontreal.ca (D.S.); 4Research Centre, Sainte-Justine University Health Center, Departments of Kinesiology, Université de Montréal, Montreal, QC H3T 1C5, Canada; maxime.caru@montreal.ca (M.C.); daniel.curnier@umontreal.ca (D.C.)

**Keywords:** childhood leukemia survivor, HDL-cholesterol, nutrition, adolescent, young adults

## Abstract

Survivors of childhood acute lymphoblastic leukemia (cALL) are at high risk of developing dyslipidemia, including low HDL-cholesterol (HDL-C). This study aimed to examine the associations between food/nutrient intake and the levels of HDL-C in a cohort of children and young adult survivors of cALL. Eligible participants (*n* = 241) were survivors of cALL (49.4% boys; median age: 21.7 years old) recruited as part of the PETALE study. Nutritional data were collected using a validated food frequency questionnaire. Fasting blood was used to determine participants’ lipid profile. Multivariable logistic regression models were fitted to evaluate the associations between intakes of macro- and micronutrients and food groups and plasma lipids. We found that 41.3% of cALL survivors had at least one abnormal lipid value. Specifically, 12.2% had high triglycerides, 17.4% high LDL-cholesterol, and 23.1% low HDL-C. Low HDL-C was inversely associated with high intake (third vs. first tertile) of several nutrients: proteins (OR: 0.27, 95% CI: 0.08–0.92), zinc (OR: 0.26, 95% CI: 0.08–0.84), copper (OR: 0.34, 95% CI: 0.12–0.99), selenium (OR: 0.17, 95% CI: 0.05–0.59), niacin (OR: 0.25, 95% CI: 0.08–0.84), riboflavin (OR: 0.31, 95% CI: 0.12–0.76) and vitamin B12 (OR: 0.35, 95% CI: 0.13–0.90). High meat consumption was also inversely associated (OR: 0.28, 95% CI: 0.09–0.83) with low HDL-C while fast food was positively associated (OR: 2.41, 95% CI: 1.03–5.63) with low HDL-C. The role of nutrition in the development of dyslipidemia after cancer treatment needs further investigation.

## 1. Introduction

Acute lymphoblastic leukemia (ALL) accounts for approximately one fourth of all childhood malignancies [1]. Cure rates now exceed 90%, which allows a growing number of childhood survivors to reach adulthood [1]. However, survivors may face severe long-term sequelae years after the end of treatments [2,3]. In particular, studies on childhood ALL (cALL) survivors have reported a high prevalence of the typical components of the metabolic syndrome, namely obesity [4], hypertension [5], glucose intolerance [6] and dyslipidemia [7]. It is important to note that cALL survivors are at higher risk of developing low levels of HDL-C [8,9,10]. Alterations in HDL particle composition in lipids and proteins were also found in this population [11,12], suggesting defects in functionality. HDL particles are known for their ability to promote reverse cholesterol transport, but their functions are more extensive in view of their anti-atherosclerotic properties such as antioxidant [13,14], anti-apoptotic [15], anti-inflammatory [16], anti-hypertensive [17] and anti-thrombotic [18] roles.

In the general population, a typical Western dietary pattern, characterized by a higher consumption of fried foods and sweetened beverages, was associated with the incidence of the metabolic syndrome components and of atherosclerosis [19,20,21]. Conversely, plant-based diets and whole grains can be protective against insulin resistance, dyslipidemia and hypertension [22,23,24,25,26]. Similarly, several key micronutrients (polyphenols, vitamins and minerals: e.g., vitamins A, C, E and selenium) exhibited anti-inflammatory and antioxidative properties [22,27]. Moreover, like the general population, cALL survivors do not respect dietary guidelines [28,29]. Thus, one can understand the relationship between inadequate nutrition and the increased risk of developing long-term sequelae by cALL survivors [28,30,31], whereas the inclusion of specific nutrients or dietary patterns was found to be protective [30,31]. In cALL survivors, greater adherence to a Mediterranean diet was associated with lower visceral adiposity, subcutaneous adiposity, waist circumference, and BMI [30]. Likewise, lower dietary quality, measured with the Healthy Eating Index-2005, was associated with higher body fat in survivors of childhood cancer [31]. Adherence to a Mediterranean diet improves HDL efflux capacity, antioxidative status and vasoprotective capacity in the general population [32]. In addition, associations were found between intake of folate and magnesium and HDL-C, HDL2, HDL3 and apolipoprotein (Apo)-AI [33]. However, the influence of dietary intakes on HDL-C levels in cALL survivors remains unknown. The major aim of this study was to study the associations between macro/micronutrient intake and the presence of low HDL-C in the PETALE cohort comprising of children and young adult survivors of cALL in the Province of Quebec, Canada.

## 2. Materials and Methods

### 2.1. Study Population

Participants were recruited between January 2013 and December 2016 as part of the PETALE study at Sainte-Justine University Health Center in Montreal (SJUHC) [34]. The PETALE study was established for investigating long-term sequelae in cALL survivors, namely cardiotoxicity, cardiometabolic complications, neurocognitive problems, bone morbidity and quality of life outcomes [34]. Participants (*n* = 246) were of European-descent living in the Province of Quebec and were treated for ALL at SJUHC according to Dana-Farber Cancer Institution-ALL protocols 87-01 to 2005-01. Participants were classified according to the types (precursor B-cell and precursor T-cell) and subtypes of ALL. The latter were grouped into 4 categories: 1. Hyperdiploidy, 2. Pre-B or pre-T with recurrent genetic abnormalities, 3. Hypodiploidy, 4. Not otherwise specified (NOS).

The Institutional Review Board of SJHUC approved the study and investigations were carried out in accordance with the principles of the Declaration of Helsinki. Written informed consent was obtained from study participants and/or parents/guardians.

### 2.2. Anthropometric Assessment

At visit, participants underwent anthropometric evaluations. Weight and height were measured with calibrated scale and height gauge. Waist circumference was measured in the horizontal plane of the superior border of the iliac crest as recommended by the National Cholesterol Education Program Third Adult Treatment Panel (NCEP ATPIII) [35]. Waist-to-height ratio (WHtR, waist circumference (cm)/height (cm)) and body mass index (BMI, weight (kg)/height (m)^2^) were calculated.

### 2.3. Biochemical Assessment

Overnight fasting blood samples were collected in tubes containing 1 g/L EDTA and were kept on ice until centrifugation. Plasma was separated within 45 min of collection and stored at −80 °C until analysis. Fasting insulin (pmol/L), glucose (mmol/L), total cholesterol (TC, mmol/L), triglycerides (mmol/L) and HDL-C (mmol/L) were measured as described previously [34]. Low-density lipoprotein-cholesterol (LDL-C, mmol/L) was calculated using the Friedewald formula [36]. For adults, values of HDL-C <1.03 mmol/L in men and <1.3 mmol/L in women were categorized as low [37]. In children, HDL-C values were classified as optimal, borderline or abnormal according to the Guidelines from the National Heart, Lung, and Blood Institute for gender and age group [38]. The homeostasis model assessment insulin resistance (HOMA-IR) was calculated using the formula: (insulin (mIU/L) × glucose (mmol/L)/22.5). Reference values for age and gender are described in Appendix A.

### 2.4. Assessment of Dietary Intake

Dietary intake was collected using a validated food frequency questionnaire (FFQ) comprising 190 items [39]. Two registered nutritionists alternately administered the FFQ to participants using measuring cups to facilitate portion size estimation. Participants were surveyed on the frequency of each item consumed during the past two months. Additional open- and closed-ended questions were asked in order to complete participants’ reported food intake, including intake of dietary supplements. A validated in-house-built nutrient calculation tool was used to evaluate nutrient intakes derived from the FFQ [39]. Energy, macronutrients and micronutrients daily intakes were calculated for all the participants. Furthermore, items from the FFQ were classified into food groups: meat, fish and seafood, dairy, fat, vegetables, legumes, fruits and fast food (Appendix A). Food groups were also classified into subgroups: white meat, red meat, processed meat, unsweetened dairy products, sweetened dairy products, nuts and seeds, refined grains with added sugar, refined grains with no added sugar and whole grains. We calculated daily intake of selected nutrients per 1000 kcal of energy (nutrient density): proteins, selenium, copper, zinc, phosphorus, riboflavin and niacin. Mean intake of nutrients was also calculated per kg of body weight.

### 2.5. Assessment of Physical Activity

Daily moderate to vigorous physical activities were measured with the Minnesota Leisure Time Physical Activity Questionnaire [40,41] and the Tecumseh Self-Administered Occupational Physical Activity Questionnaire [42]. An experienced exercise physiologist conducted the interviews. The exercise physiologist first read the 20 sports included in the questionnaire and then guided the participants to recall any other sports or leisure physical activities they might have practiced in the last three months. Clarifications on frequency, duration and intensity of the activities were asked. A metabolic equivalent value (MET) from the Compendiums of Physical Activity for Adults and for Youth [43] was used to quantify the intensity of each activity. All activities with a MET value ≥3 were considered of moderate-to-vigorous intensity [43]. Total of daily minutes of moderate-to-vigorous leisure physical activities was then calculated.

### 2.6. Assessment of Estimated Energy Requirement, Energy Balance and Recommended Dietary Allowance

The definition of estimated energy requirement (EER) is the average dietary energy intake that is predicted to maintain energy balance in healthy, normal weight individuals of a defined age, gender, weight, height and level of physical activity consistent with good health [44]. The equation to calculates the EER differs between children/adolescents and adults. In addition to the total energy expenditure, the equation considers the needed energy for growth in children and adolescents (Appendix A). The EER definition includes a physical activity coefficient that varies for gender and age and is categorized as sedentary, low active, active and very active (Appendix A). Energy balance was calculated as the difference between participant’s total daily energy intake and the EER. Recommended dietary allowance (RDA) of nutrients was determined for each participant according to their age group and gender using the recommendations (g/day) published by Health Canada [44]. Mean daily intake of nutrients was expressed as percentage of RDA.

### 2.7. Statistical Analysis

Wilcoxon signed-rank tests were used to compare energy and nutrient intake between cALL survivors with normal and low HDL-C. After an univariate analysis, multivariable logistic regression models were fitted to evaluate the associations between intakes of macro- and micronutrients and having low HDL-C. Tertiles or medians of the macro-micronutrients were examined for association with low HDL-C after accounting for potential confounding from: BMI, age at diagnosis, sex, total energy intake and total of daily minutes of moderate-to-vigorous leisure physical activities. To enhance model fit, the square of the age variable was also included. Separate models were fit to examine associations between individual macro-micronutrients. For food groups, a single model that included all food groups (after accounting for potential confounders) was fit. Model adequacy in terms of its specification and goodness-of-fit was examined using standard methods. Chi-square tests were used to evaluate the relationships between ALL types/subtypes and HDL-C outcome. A *p* value inferior to 0.05 was considered statistically significant. Analyses were carried out using the STATA 10.1 version.

## 3. Results

### 3.1. Descriptive Statistics

Demographic and clinical characteristics of participants are summarized in Table 1. Five participants were excluded because of incomplete nutritional data. A total of 241 participants were included in the analyses (49.4% men) comprising of 156 adults (≥18 years old) and 85 children (<18 years old). Median age at diagnosis was 4.7 years and median time since diagnosis was 15.4 years. Precursor B-cell and NOS were the most common type and subtype of ALL, respectively. Types and subtypes of ALL were found to have no influence on HDL-C outcome (data not shown). We found that 41.1% of participants had dyslipidemia defined by at least one abnormal lipid value. Specifically, 12.0% had high triglycerides, 17.4% high LDL-cholesterol and 22.8% had low HDL-C. However, the group median for HDL-C was 1.30 mmol/L (range: 1.27 to 1.34 mmol/L) that is classified as normal (Table 1). WHtR has been suggested as a better predictor than waist circumference for the metabolic syndrome and dyslipidemia [45,46]. A WHtR superior to 0.5 was associated with higher cardiovascular risk for adults and children of both genders [46]. Here, we found that 53.4% of the participants had WHtR >0.5, and that the prevalence was higher in adults than children (61.3% vs. 39.3%, respectively). A high BMI (adults: >30 kg/m^2^; children: ≥97th percentile) was found in 13.7% of the participants (16.7% in adults). Moreover, the median time of moderate-to-vigorous activity was 20.6 min per day and was higher in children than in adults (28.9 vs. 15.7 min/day) (Table 1).

Participants’ daily intake of energy, macro- and micronutrient is presented in Table 2. Between the upper (tertile 3) and the lower tertile (tertile 1), the median of macronutrient intake was higher by 72% (carbohydrates) to 101% (dietary fibers). A higher range of intakes was observed for micronutrients, the differences between the third and first tertiles reaching 180% (vitamin C and D).

Then we expressed participants’ intake of specific nutrients as percentage of RDA (proteins, selenium, copper, zinc, phosphorus, riboflavin and niacin) (Figure 1). For riboflavin, selenium and proteins, the intakes of all the participants were above 100% of the RDA. Yet, few participants did not meet 100% of the RDA for: niacin (*n* = 7), zinc (*n* = 7), copper (*n* = 3) and phosphorus (*n* = 11). Overall, it appears that most of the participants had intakes superior to the RDA for the nutrients listed.

### 3.2. Association between Macro- and Micronutrient Intake and Low HDL-C

The multivariable logistic regression analysis showed that higher intakes in several nutrients (third vs. first tertile) were inversely associated with low HDL-C presentation (Table 3, Table 4 and Table 5). These were: proteins (odd ratio (OR): 0.29, 95% CI: 0.08–1.00, *p* = 0.05), zinc (OR: 0.26, 95% CI: 0.08–0.84, *p* < 0.05), copper (OR: 0.27, 95% CI: 0.09–0.81, *p* < 0.05), selenium (OR: 0.16, 95% CI: 0.05–0.59, *p* < 0.01), niacin (OR: 0.26, 95% CI: 0.08–0.88, *p* < 0.05), and riboflavin (OR: 0.25, 95% CI: 0.07–0.86, *p* < 0.05). Of note, participants’ level of physical activity, gender and the age at the interview were not associated with the low HDL-C outcome, whereas higher BMI and high daily energy intake were associated with low-HDL-C in all models (data not shown, but available on request) (Table 3, Table 4 and Table 5).

Next, we analyzed the associations between intakes of mean daily portions of food groups (meat, fish, dairy, fats, vegetables, legumes, fruits and fast food) and low HDL-C (Table 6). A higher intake of foods classified in the meat group (tertile 3 vs. tertile 1) was inversely associated with having low HDL-C (OR: 0.28, 95% CI: 0.09–0.83, *p* = 0.022). On the contrary, a higher intake of fast food (tertile 2 vs. tertile 1) was associated with higher risk of having low HDL-C (OR: 2.41, 95% CI: 1.03–5.63, *p* = 0.043). No associations between the food subgroups (white meat, red meat, processed meat, nuts and seeds, unsweetened dairy products, sweetened dairy products, refined grains with no added sugar, refined grains with added sugar and whole grains) and low HDL-C were observed (data not shown).

### 3.3. Relationship between Mean Density Intake/Daily Intake and HDL-C Status

We compared mean density intake (g/1000 kcal) between cALL survivors with and without low HDL-C for macronutrients and micronutrients for which associations were established using logistic regressions. There were no statistically significant differences in energy and macronutrient intake per 1000 kcal between the two groups (Figure 2).

Nonetheless, mean density intake of selenium, copper, zinc and niacin were lower in participants with low HDL-C than those with normal values (Figure 3). No difference was found for riboflavin and phosphorus (Figure 3).

Mean daily intake of proteins (g/kg body weight) was lower (1.63 ± 0.61) in cALL survivors with low HDL-C compared to survivors with normal HDL-C levels (1.8 ± 0.55) (Figure 4). However, there was no statistically significant difference between the two groups for energy and other macronutrients (g/kg body weight) (Figure 4).

Mean daily intakes of five micronutrients were lower in participants with low HDL-C: selenium (2.39 ± 0.87 vs. 2.07 ± 0.86 mg/kg body weight), copper (0.032 ± 0.012 vs. 0.028 ± 0.017 mg/kg body weight), zinc (0.27 ± 0.09 vs. 0.23 ± 0.09 mg/kg body weight), riboflavin (0.05 ± 0.02 vs. 0.04 ± 0.01 mg/kg body weight) and niacin (0.45 ± 0.16 vs. 0.40 ± 0.17 mg/kg body weight) (Figure 5).

## 4. Discussion

The main findings of this study are the identification of an inverse association between high dietary intake of specific macro- and micronutrients and HDL-C levels in cALL survivors: proteins, selenium, zinc, copper, riboflavin and niacin. We also found that higher consumption of meat (including red and white meat), as a food group, was associated with having normal HDL-C, while fast food was associated with low HDL-C.

To date, few studies have evaluated the relationship between diet and cardiometabolic outcomes in pediatric cancer survivors, and none has specifically examined the associations with HDL-C. Poor adherence to dietary guidelines among adult survivors of cALL has been reflected by excess consumption of sugar, fat and insufficient intakes of whole grains, vegetables and fruits [28,29]. Inadequate eating habits were also observed in our study. For example, average daily serving of fast food was >0.5 portion, serving of whole grains was <1 portion, and daily intake of fruits and vegetables were 2.1 and 3.1 portions respectively. In the general population, intake of specific nutrients such as proteins, folate and magnesium has been associated with higher blood levels of HDL-C [33,47]. Alcohol intake and mystic acid (14:0) were also associated to higher HDL-C levels, while dietary carbohydrates and iron were related to lower HDL-C [33]. A crossover clinical trial showed that a diet rich in complex carbohydrates, in replacement of fat (mainly monounsaturated fat), did not impact HDL-C concentrations [48]. However, it did increase the catabolic rate of several proteins on HDL particles such as apolipoprotein (Apo)-A1, Apo-A2 and Apo-E [48]. Moreover, a meta-analysis examining the effects of low-carbohydrate diets on body weight and cardiovascular risk showed that participants on a low-carbohydrate diet, compared with those on a low-fat diet, had a greater increase of HDL-C and of weight loss [49]. However, we did not find an association between carbohydrate or lipid intake, as protein was the only macronutrient associated with HDL-C.

In our study, a high BMI was associated with a low HDL-C. Adipose tissue is considered as an endocrine organ that releases pro-inflammatory adipokines, including plasminogen activator inhibitor-1 (PAI-1), TNF-α, interleukin-6 (IL-6), resistin, leptin and adiponectin [50]. The secretion of these mediators alters body weight homoeostasis and contributes to the development of insulin resistance, diabetes and dyslipidemia [51]. The relationship between abdominal obesity and low HDL-C has been demonstrated is several studies [52,53,54,55]. Furthermore, in our study, moderate-to-vigorous physical activity was not associated with HDL-C in the logistic regression models. This finding is contrary to many reports showing that physical activity positively impacts lipid profile [56,57] and particularly HDL-C [57,58,59,60]. However, data from the England and Scottish Health Survey (cohort of 37,059 adults) showed no statistically significant association between physical inactivity and having low HDL-C [61].

We found that higher protein intake was inversely associated with low HDL-C. However, protein intake was superior to the RDA for both participants with low and normal HDL-C. The RDA of 0.8 g protein/kg body weight represents the minimum protein consumption required to avoid a deficiency that could lead to the progressive loss of muscle mass in healthy adults [62,63]. Previous studies reported comparable findings in the general population. In a study using data from the NHANES 2001–2010 (*n* = 23,876 adults ≥19 years old), usual protein intake varied across deciles from 0.69 ± 0.004 to 1.51 ± 0.009 g/kg body weight and a diet higher in protein was associated with higher HDL-C levels [47]. The mechanisms by which protein intake could elevate HDL-C concentrations are still unknown, but it has been proposed that their properties, other than energy content, could be responsible for their beneficial effect [47].

In our study, only two food groups were found to be associated with the HDL-C outcome: fast food and meat. Similarly, in a prospective study that took place over a period of 20 years in young adults (*n* = 5115; 45.2 ± 3.6 years old), participants in the highest quartile of fast food intake, compared to the lowest quartile, were on average 5.7 kg heavier (95%CI: 2.1–9.2; *p* = 0.002), had a higher waist circumference of 5.3 cm (95%CI: 2.8–7.9; *p* < 0.001) and had lower HDL-C by 0.14 mmol/L (95%CI: −8.3–2.6; *p* < 0.001) [64]. Meat provides high-quality proteins, essential amino acids and micronutrients such as iron, vitamins A and B12, zinc, selenium and folic acid. However, lines of evidence suggested that diets high in red (beef, pork, lamb and veal) and processed meat (smoked, cured or preserved) are associated with the development of cardiovascular diseases [65,66], but this remains controversial [67,68,69,70]. The impact of weekly consumption of 200 and 500 g of unprocessed, lean red meat in a Mediterranean diet pattern was assessed in a two 5-week randomized, crossover, controlled trial. [71]. At the end of the intervention, there was no difference in the decrease of HDL-C between the two diets [71]. The results of a meta-analysis, including randomized controlled trials, showed that the consumption of ≥0.5 servings of red meat/day do not influence HDL-C [72]. In our study, participants consumed on average 0.44 serving of red meat per day (33 g/per day or 231 g/per week). The meat food group included red, white and processed meat. Stratifying the analysis into the three meat sub-groups did not reveal any significant association or trend, most likely because of the lack of power. Additional studies are needed in populations of cALL survivors to confirm and understand the association between meat and HDL-C.

A network meta-analysis ranked 10 major food groups (refined grains, whole grains, fruits and vegetables, nuts, legumes, eggs, dairy, red meat, fish, and sugar-sweetened beverages) according to their effects on cardiometabolic outcomes including HDL-C [73]. Fish was ranked first at increasing HDL-C before nuts, eggs, red meat, refined grains and whole grains that were also found effective [73]. Legumes were ranked last after sugar-sweetened beverages. While these results are in line with our findings for the meat food group, we did not observe associations between the consumption of fish, nuts, whole grains and refined grains and HDL-C. The small sample size might explain the lack of associations. Similarly, in our study, the legume food group composed of plant proteins (legumes and tofu) was not associated with HDL-C in the logistic regression model. This could be partially explained by the fact that the average daily consumption of legumes was very low (group median of 0.03 portion/day).

Selenium, zinc, copper, riboflavin and niacin are minerals and vitamins found mostly in foods rich in proteins. Cereal products, legumes, nuts, meat, fish, dairy and eggs are the main sources for these micronutrients. For four of these five nutrients, the mean density intake (g/1000 kcal) was significantly lower in cALL survivors with low HDL-C. However, intakes were greater than or very close to 100% of the RDA for all the participants. Similarly, the American national survey NHANES (2007–2010) reported a low prevalence of intake inadequacy for all B vitamins and for several minerals including copper, phosphorus, selenium and zinc [74].

Some of the main functions of these micronutrients are related to metabolism, gene transcription and oxidative balance [75]. Selenium, a trace metal, is known for its antioxidant capacity that is attributed to its integration in selenoproteins such as glutathione peroxidase [76]. Here, we found a positive relationship between selenium intake and HDL-C. To our knowledge, this has not been investigated elsewhere. However, the impact of selenium blood levels or supplementation on HDL-C is conflicting: a study reported a positive association [77] whereas others did not find any relationship [78,79,80].

Zinc is a structural component of proteins and participates in several cellular functions including cell proliferation and differentiation, RNA and DNA synthesis, stabilization of cell membranes, redox regulation and apoptosis [81]. A meta-analysis has highlighted that zinc supplementation in humans reduces HDL-C levels by 0.05 mmol/L (95% CI: −0.74–4.98) compared to placebo, but without reaching statistical significance (*p* = 0.15) [82]. However, in metabolically non-healthy participants, zinc supplementation increased HDL-C (0.16 mmol/L, 95%CI: 2.38–9.92, *p* < 0.05) [82]. Therefore, it appears that the positive relationship between zinc intake and HDL-C observed in our study is comparable to previous findings in the metabolically unhealthy.

Copper is essential for various enzymes that catalyze redox reactions, including superoxide dismutase [83]. Copper supplementation (5 mg/day) for 45 days in 73 hypercholesterolemic patients (26–46 years old) decreased total cholesterol, LDL-C and triglycerides and slightly increased HDL-C [84]. However, in a study including healthy men with adequate copper status, copper supplementation did not affect lipid profiles [85]. Since survivors of childhood cancer are at risk for metabolic alterations [4,5,6,7], the impact of copper intake on HDL-C might be of greater amplitude that for healthy populations.

Riboflavin, or vitamin B2, is essential for haematological, neurological, cardiovascular, and endocrine system functioning because of its role in mitochondrial and energy metabolism [86]. A study using data from the Korean National Health and Nutrition Examination Survey 2015–2016 (*n* = 6062; ≥19 years old) had findings in line with ours: poor riboflavin intake (<1.5 mg/day) in men was associated with low HDL-C (OR: 1.362, 95% CI: 1.017–1.824, *p* = 0.038) [86]. However, in our cohort, riboflavin intakes of all participants were superior to the RDA.

Niacin is a collective term for nicotinic acid and its derivatives, such as nicotinamide. Nicotinic acid is a soluble vitamin that belongs to the B vitamin group [87]. Pharmacological doses of nicotinic acid impact lipid profile by increasing HDL-C and reducing triglycerides, LDL-C and lipoprotein (a) [88]. Moreover, a lower dietary intake of the vitamins niacin, riboflavin thiamine, B6 and vitamin C has been associated with an increased severity of coronary disease [89]. However, the relationship between niacin dietary intake and HDL-C has not been assessed elsewhere, to our knowledge.

A strength of our study is that it is the first to examine the influence of dietary intakes on HDL-C levels in cALL survivors, a population at increased risk of dyslipidemia. The use of a validated FFQ tailored for our targeted population is another strength. One limitation is having solely examined the associations between nutrient intake and HDL-C levels without investigating the impact on HDL composition and functionality. HDL-C measurements are important to assess cardiovascular risk [90], but it appears that lipoprotein composition rather than concentration better predicts cardiovascular events [91,92]. Next, FFQs are tools commonly used to estimate usual food intake in epidemiological studies in order to examine associations between diet and diseases [93]. It is possible that dietary intakes were over- or underestimated even with the use of a validated FFQ. In addition, although we included relevant covariates in our multivariable logistic regression analysis to avoid confounding effects, variables that were not tested (e.g., smoking and alcohol) could have had an effect. Moreover, given the cross-sectional measurements, causality between nutrient intakes and HDL-C levels cannot be proven. In addition, because of the limited sample size, we were not able to further stratify the analyses to investigate associations with food sub-groups and hence better explain our findings. Finally, data were not compared to a control population without history of cancer. Thus, the study outcomes are purely descriptive and need to be validated in other cohorts.

## 5. Conclusions

Our study supports a potential beneficial effect of dietary proteins, meat, selenium, zinc, copper, riboflavin, and niacin and a deleterious effect of fast food on HDL-C in cALL survivors. This preliminary groundwork provides fundamentals that contribute to our understanding of the role of nutrition in the development of dyslipidemia in this population. Based on these results, we propose that a diet rich in high-quality lean proteins and in key micronutrients could contribute to protect cALL survivors against having low HDL-C. This study emphasizes the role of nutrition in the development of dyslipidemia after cancer treatment. Nutritional strategies could be a valuable approach to preventing long-term cardiometabolic complications in cALL survivors.

## Figures and Tables

**Figure 1 nutrients-11-02977-f001:**
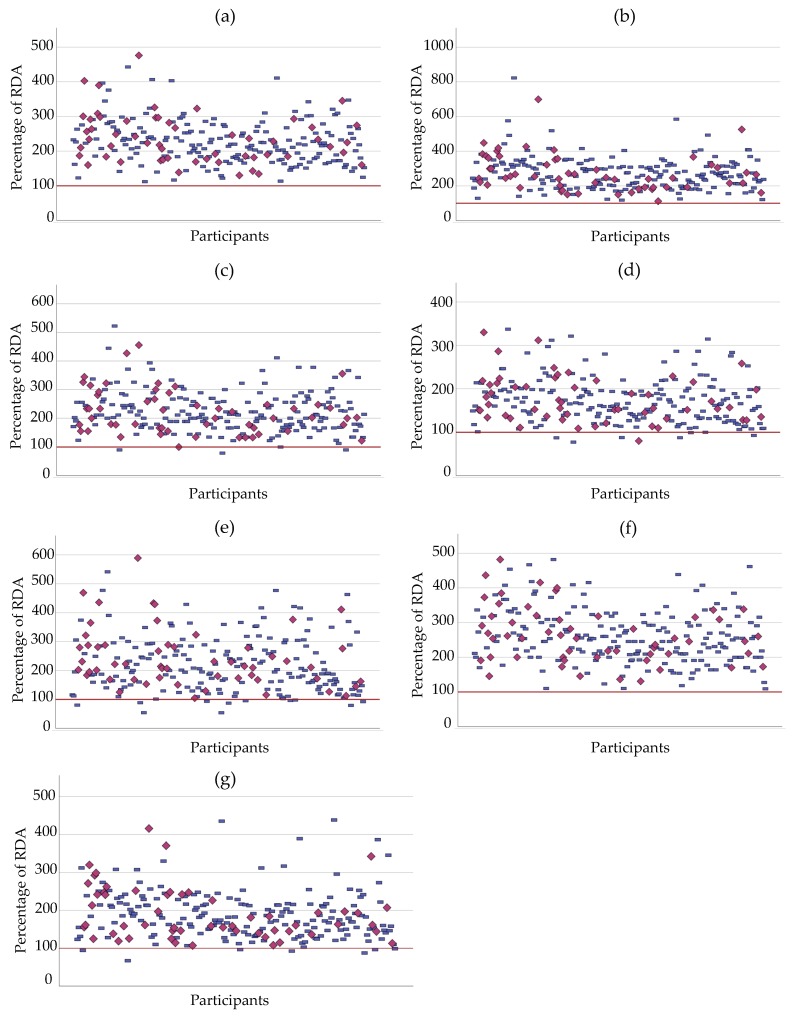
Intakes in proteins, minerals and vitamins of cALL survivors expressed as percentage of RDA (y axis) for each participant (x axis). (**a**) Proteins; (**b**) Selenium; (**c**) Copper; (**d**) Zinc; (**e**) Phosphorus; (**f**) Riboflavin; (**g**) Niacin. HDL-C, high density lipoprotein-cholesterol; RDA, recommended dietary allowance. Blue rectangle = Normal HDL; red diamond = Low HDL-C.

**Figure 2 nutrients-11-02977-f002:**
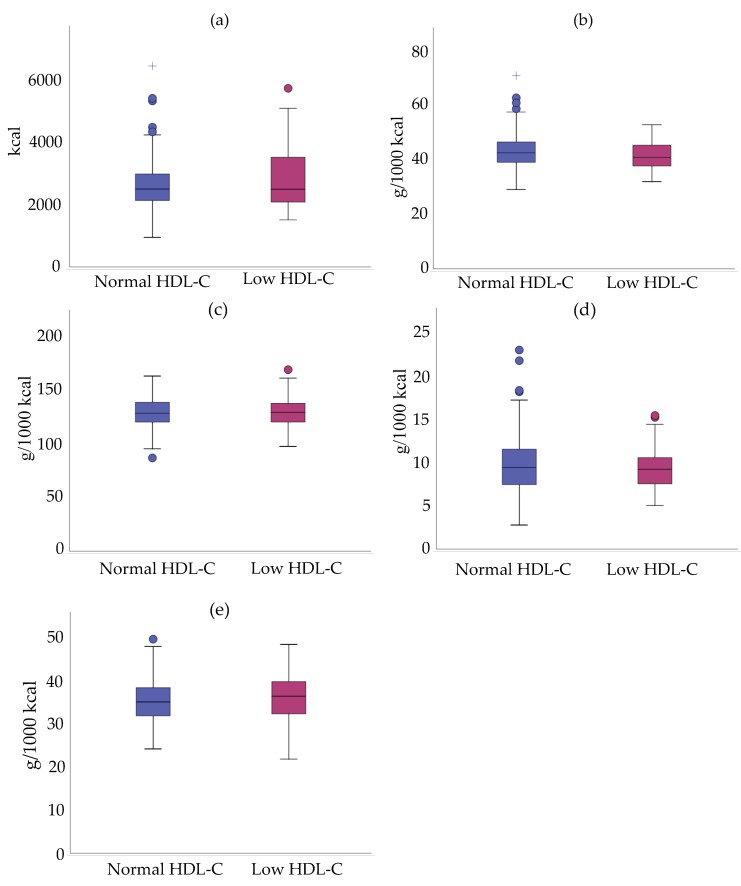
Intakes in energy and macronutrients per 1000 kcal of cALL survivors with low and normal HDL-C. (**a**) Energy intake; (**b**) Proteins; (**c**) Carbohydrates; (**d**) Dietary fibers; (**e**) Lipids. Data are presented as median ± interquartile range.

**Figure 3 nutrients-11-02977-f003:**
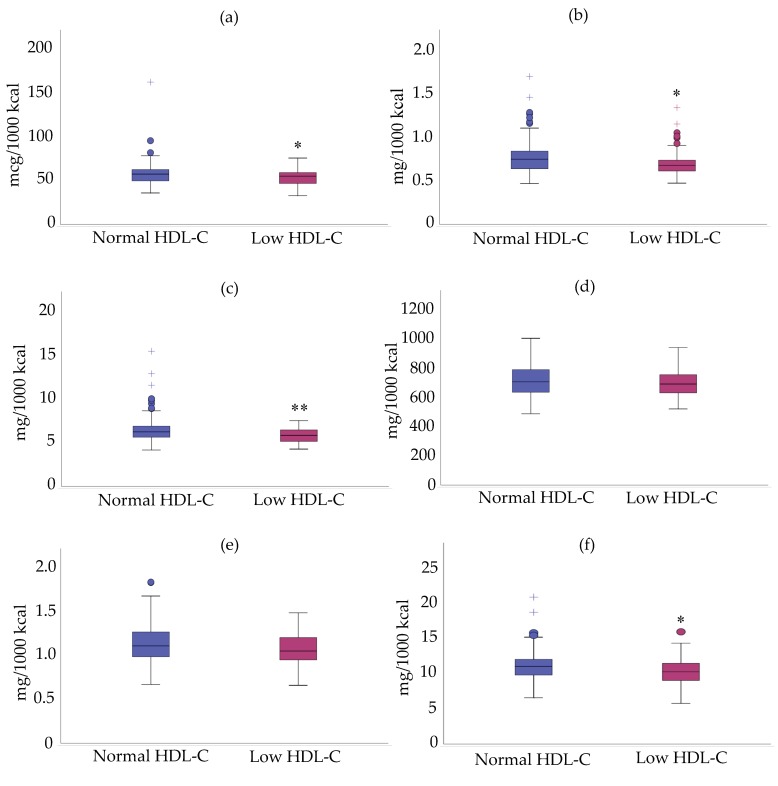
Intakes in micronutrients per 1000 kcal of cALL survivors with low and normal HDL-C. (**a**) Selenium; (**b**) Copper; (**c**) Zinc; (**d**) Phosphorus; (**e**) Riboflavin; (**f**) Niacin. * *p* < 0.05 versus ALL survivors with normal HDL-C. Data are presented as median ± interquartile range. ** *p* < 0.01 versus ALL survivors with normal HDL-C.

**Figure 4 nutrients-11-02977-f004:**
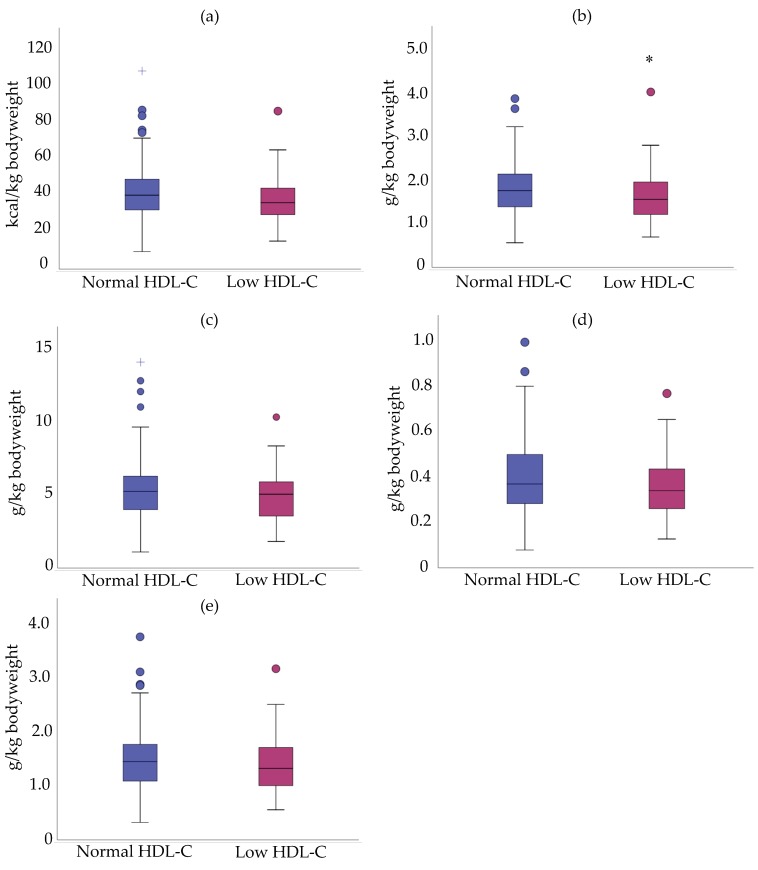
Intakes in energy (kcal/kg bodyweight) and macronutrients (g/kg bodyweight) of cALL survivors with low and normal HDL-C. (**a**) Energy intake; (**b**) Proteins; (**c**) Carbohydrates; (**d**) Dietary fibers; (**e**) Lipids. Data are presented as median ± interquartile range. * *p* < 0.05 versus ALL survivors with normal HDL-C.

**Figure 5 nutrients-11-02977-f005:**
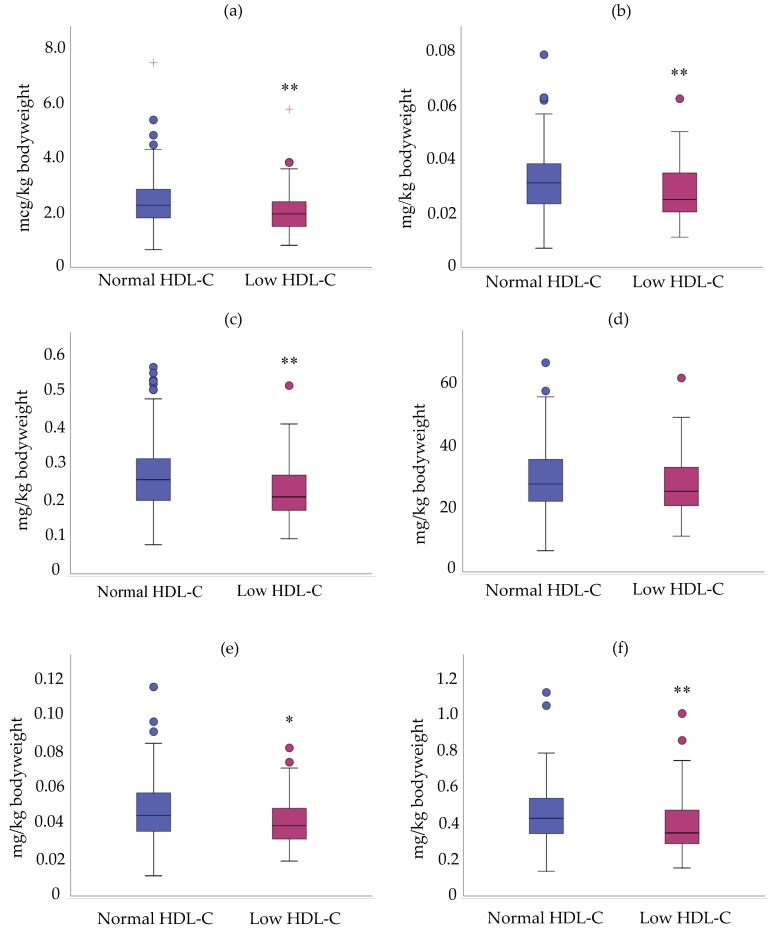
Intakes in micronutrients (mg/kg bodyweight) of cALL survivors with low and normal HDL-C. (**a**) Selenium; (**b**) Copper; (**c**) Zinc; (**d**) Phosphorus; (**e**) Riboflavin; (**f**) Niacin. * *p* < 0.05 versus ALL survivors with normal HDL-C. Data are presented as median ± interquartile range. ** *p* < 0.01 versus ALL survivors with normal HDL-C.

**Table 1 nutrients-11-02977-t001:** Demographic and clinical characteristics of participants.

	Total *n* = 241	Adults *n* = 156	Children *n* = 85
	Median (range or interquartile range ^1^)		
Age at visit, years (range)	21.3 (8.5–40.9)	24.6 (18.0–40.9)	16.2 (8.5–17.9)
Age at cancer diagnosis, years (range)	4.7 (0.9–18.0)	6.5 (0.9–18.0)	3.5 (1.3–10.9)
Time since end of treatment, years (range)	12.9 (3.3–26.1)	16.11 (3.9–26.1)	9.60 (3.3–13.4)
Gender (male, %)	49.4	57.3	49.4
ALL types ^2^			
Pre-B ALL (*n*, %)	208 (88.5)	128 (84.8)	80 (95.2)
Pre-T ALL (*n*, %)	27 (11.5)	23 (15.2)	4 (4.8)
ALL subtypes ^3^			
Hyperdiploidy (*n*, %)	33 (21.6)	18 (19.8)	15 (24.2)
Pre-B or Pre-T with recurrent genetic abnormalities (*n*, %)	52 (34.0)	28 (30.8)	24 (38.7)
Hypodiploidy (*n*, %)	4 (2.6)	3 (3.3)	1 (1.6)
Others (NOS) (*n*, %)	64 (41.8)	42 (46.1)	22 (35.5)
Metabolic data			
Physical activity 4 (min/day)	20.6 (7.6–39.5)	15.7 (3.95–36.6)	28.9 (14.1–46.5)
Estimated energy requirement (kcal)	2328 (2011–2721)	2373 (2039–2707)	2224 (1964–2834)
Energy balance (kcal)	142 (−213–649)	133 (−224–644)	148 (−172–679)
BMI (kg/m^2^)	23.5 (20.9–26.1)	24.3 (21.7–27.4)	21.8 (19.2–24.1)
Waist-to-height ratio	0.50 (0.46–0.55)	0.51 (0.48–0.58)	0.49 (0.45–0.52)
Glucose (mmol/L)	5.0 (4.8–5.3)	5.0 (4.8–5.4)	5.0 (4.8–5.3)
Insulin (pmol/L)	53.3 (37.6–75.8)	50.1 (35.2–68.6)	58.1 (42.9–81.1)
HOMA-IR	1.7 (1.2–2.5)	1.7 (1.1–2.3)	1.9 (1.3–2.7)
Total cholesterol (mmol/L)	4.37 (3.87–5.01)	4.59 (4.10–5.15)	4.18 (3.61–4.63)
Triglyceride (mmol/L)	0.91 (0.66–1.25)	0.97 (0.72–1.38)	0.82 (0.62–1.07)
LDL-C (mmol/L)	2.57 (2.13–3.16)	2.73 (2.22–3.32)	2.36 (2.04–2.82)
HDL-C (mmol/L)	1.30 (1.12–1.49)	1.31 (1.13–1.52)	1.29 (1.09–1.45)

^1^ Interquartile range: 25th and 75th percentiles. ^2^ ALL types were not available for six participants. ^3^ ALL subtypes were not available for 88 participants out of 241. ^4^ Daily minutes of moderate-to-vigorous leisure physical activities. ALL, acute lymphoblastic leukemia; NOS, not otherwise specified; BMI, body mass index; HOMA-IR, homeostasis model assessment-insulin resistance; LDL-C, low-density lipoprotein-cholesterol; HDL-C, high-density lipoprotein-cholesterol.

**Table 2 nutrients-11-02977-t002:** Participants’ intake of energy, macro- and micronutrients.

	Total	Tertile 1	Tertile 2	Tertile 3
	Median (interquartile range ^1^)			
Energy intake (kcal)	2512 (2143–3077)	1897	2511	3398
Macronutrients				
Proteins (g)	109 (87.3–131)	80.4	108.9	144.0
Carbohydrates (g)	318 (267–391)	247.8	317.9	426.4
Dietary fibers (g)	23.8 (19.0–29.6)	16.6	24.0	33.4
Lipids (g)	90.2 (71.6–114.8)	63.0	90.2	123.1
Omega-6 (g)	4.4 (3.1–6.5)	2.6	4.4	7.9
Omega-3 (g)	0.8 (0.6–1.4)	0.5	0.8	1.8
Ratio ω-6/ω-3	5.3 (4.0–6.9)	3.6	5.4	7.5
Micronutrients				
Calcium (mg)	1349 (1047–1782)	931	1349	1993
Iron (mg)	16.3 (13.6–21.5)	12.7	16.3	23.5
Magnesium (mg)	405 (342–497)	311	406	547
Phosphorus (mg)	1810 (1452–2192)	1351	1810	2445
Potassium (mg)	3975 (3337–4763)	3028	3975	5202
Sodium (mg)	3428 (2673–4522)	2372	3428	4996
Zinc (mg)	15.6 (12.4–19.5)	11.6	15.6	21.2
Copper (mg)	1.9 (1.5–2.3)	1.4	1.8	2.4
Manganese (mg)	4.1 (3.2–5.2)	2.8	4.1	5.8
Selenium (mcg)	139 (112–176)	102	139	192
Retinol (mcg)	469 (330–681)	286	469	744
Folic acid (mcg)	628 (505–781)	445	627	871
Niacin (mg)	26.5 (21.0–33.6)	19.0	26.2	35.9
Riboflavin (mg)	2.8 (2.3–3.5)	2.1	2.8	3.8
Thiamine (mg)	2.2 (1.8–2.8)	1.5	2.2	3.0
Vitamin B6 (mg)	2.4 (1.9–2.9)	1.7	2.3	3.1
Vitamin B12 (mcg)	5.7 (4.3–7.7)	3.8	5.7	8.6
Choline (mg)	283 (224–355)	201	283	389
Vitamin C (mg)	210 (136–275)	114	210	319
Vitamin D (mcg)	5.6 (4.1–8.4)	3.5	5.6	9.80
Vitamin K (mcg)	122 (87.4–170)	74.1	122	204
Food groups				
Meat	1.8 (1.3–2.6)	1.0	1.8	3.0
Fish and seafood	0.3 (0.1–0.5)	0.03	0.3	0.7
Dairy	3.2 (2.1–4.7)	1.6	3.2	5.0
Fat	1.5 (1.0–2.4)	0.7	1.5	3.1
Vegetables	3.1 (2.0–4.5)	1.5	3.1	5.1
Legumes	0.03 (0.0–0.1)	0.0	0.03	0.2
Fruits	1.9 (1.2–3.1)	0.9	1.9	3.6

^1^ Interquartile range: 25th and 75th percentiles. ω, omega; g, grams; mg, milligrams; mcg, micrograms.

**Table 3 nutrients-11-02977-t003:** Association between macronutrient intake and low HDL-C in cALL survivors.

Macronutrients	Odds Ratio	95% CI	*p* Value
Proteins			
Tertile 2 vs. Tertile 1	0.300	0.12–0.74	0.009
Tertile 3 vs. Tertile 1	0.289	0.08–1.00	0.05
Carbohydrates			
Tertile 2 vs. Tertile 1	0.705	0.29–1.70	0.436
Tertile 3 vs. Tertile 1	0.612	0.17–2.19	0.450
Fats			
Tertile 2 vs. Tertile 1	0.723	0.30–1.74	0.468
Tertile 3 vs. Tertile 1	0.876	0.26–2.91	0.829
Fibers			
Tertile 2 vs. Tertile 1	0.914	0.41–2.02	0.824
Tertile 3 vs. Tertile 1	0.603	0.23–1.59	0.308
Omega-3			
Tertile 2 vs. Tertile 1	1.347	0.59–3.05	0.475
Tertile 3 vs. Tertile 1	1.002	0.40–2.53	0.995
Omega-6			
Tertile 2 vs. Tertile 1	0.897	0.39–2.10	0.800
Tertile 3 vs. Tertile 1	0.652	0.26–1.61	0.354
Ratio omega-3: omega-6			
Tertile 2 vs. Tertile 1	1.087	0.48–2.44	0.840
Tertile 3 vs. Tertile 1	1.385	0.62–3.09	0.426

HDL-C, high-density lipoprotein-cholesterol; cALL: childhood acute lymphoblastic leukemia; CI, confidence interval. Multivariable logistic regression model adjusted for BMI (kg/m^2^), age at diagnosis (years), age at diagnosis squared (years), sex (female), total energy intake (kcal) and moderate-to-vigorous physical activity (minutes per day).

**Table 4 nutrients-11-02977-t004:** Association between mineral intake and low HDL-C in cALL survivors.

Minerals	Odds Ratio	95% CI	*p* Value
Calcium			
Tertile 2 vs. Tertile 1	0.774	0.33–1.80	0.553
Tertile 3 vs. Tertile 1	0.830	0.31–2.22	0.711
Magnesium			
Tertile 2 vs. Tertile 1	0.624	0.27–1.42	0.262
Tertile 3 vs. Tertile 1	0.350	0.11–1.12	0.078
Phosphorus			
Tertile 2 vs. Tertile 1	0.362	0.15–0.88	0.024
Tertile 3 vs. Tertile 1	0.333	0.10–1.13	0.077
Potassium			
Tertile 2 vs. Tertile 1	0.754	0.32–1.79	0.523
Tertile 3 vs. Tertile 1	0.692	0.22–2.18	0.528
Sodium			
Tertile 2 vs. Tertile 1	0.382	0.15–0.97	0.044
Tertile 3 vs. Tertile 1	1.134	0.35–3.65	0.832
Iron			
Tertile 2 vs. Tertile 1	0.478	0.21–1.11	0.086
Tertile 3 vs. Tertile 1	0.395	0.12–1.27	0.118
Zinc			
Tertile 2 vs. Tertile 1	0.311	0.13–0.76	0.010
Tertile 3 vs. Tertile 1	0.257	0.08–0.84	0.025
Copper			
Tertile 2 vs. Tertile 1	0.32	0.13–0.76	0.009
Tertile 3 vs. Tertile 1	0.27	0.09–0.81	0.020
Manganese			
Tertile 2 vs. Tertile 1	0.616	0.27–1.39	0.243
Tertile 3 vs. Tertile 1	0.639	0.25–1.60	0.340
Selenium			
Tertile 2 vs. Tertile 1	0.377	0.16–0.89	0.026
Tertile 3 vs. Tertile 1	0.175	0.05–0.62	0.007

HDL-C, high-density lipoprotein cholesterol; cALL: childhood acute lymphoblastic leukemia; OR, odds ratio; CI, confidence interval. Multivariable logistic regression model adjusted for BMI (kg/m^2^), age at diagnosis (years), age at diagnosis squared (years), sex (female), total energy intake (kcal) and moderate-to-vigorous physical activity (minutes per day).

**Table 5 nutrients-11-02977-t005:** Association between vitamin intake and low HDL-C in cALL survivors.

Vitamins	Odds Ratio	95% CI	*p* Value
Retinol			
Tertile 2 vs. Tertile 1	0.639	0.28–1.47	0.291
Tertile 3 vs. Tertile 1	0.609	0.24–1.56	0.301
Alpha-carotene			
Tertile 2 vs. Tertile 1	1.444	0.66–3.16	0.356
Tertile 3 vs. Tertile 1	0.880	0.39–2.00	0.760
Beta-carotene			
Tertile 2 vs. Tertile 1	1.523	0.67–3.44	0.312
Tertile 3 vs. Tertile 1	0.887	0.37–2.15	0.790
Thiamin			
Tertile 2 vs. Tertile 1	0.634	0.27–1.51	0.302
Tertile 3 vs. Tertile 1	0.741	0.26–2.11	0.575
Riboflavin			
Tertile 2 vs. Tertile 1	0.300	0.12–0.74	0.009
Tertile 3 vs. Tertile 1	0.248	0.07–0.86	0.028
Niacin			
Tertile 2 vs. Tertile 1	0.268	0.11–0.65	0.004
Tertile 3 vs. Tertile 1	0.263	0.08–0.88	0.030
Vitamin B6			
Tertile 2 vs. Tertile 1	0.871	0.38–2.01	0.747
Tertile 3 vs. Tertile 1	0.395	0.12–1.27	0.119
Choline			
Tertile 2 vs. Tertile 1	0.480	0.20–1.16	0.104
Tertile 3 vs. Tertile 1	0.518	0.18–1.50	0.225
Folic acid			
Tertile 2 vs. Tertile 1	0.624	0.26–1.47	0.281
Tertile 3 vs. Tertile 1	0.571	0.20–1.66	0.304
Vitamin B12			
Tertile 2 vs. Tertile 1	0.713	0.31–1.63	0.424
Tertile 3 vs. Tertile 1	0.580	0.22–1.55	0.276
Vitamine C			
Tertile 2 vs. Tertile 1	0.850	0.37–1.93	0.698
Tertile 3 vs. Tertile 1	0.864	0.36–2.07	0.744
Vitamin D			
Tertile 2 vs. Tertile 1	0.713	0.32–1.60	0.414
Tertile 3 vs. Tertile 1	0.633	0.26–1.53	0.309
Vitamin K			
Tertile 2 vs. Tertile 1	1.181	0.51–2.71	0.695
Tertile 3 vs. Tertile 1	0.988	0.41–2.40	0.978

HDL-C, high-density lipoprotein-cholesterol; cALL: childhood acute lymphoblastic leukemia; OR, odds ratio; CI, confidence interval. Multivariable logistic regression model adjusted for BMI (kg/m^2^), age at diagnosis (years), age at diagnosis squared (years), sex (female), total energy intake (kcal) and moderate-to-vigorous physical activity (minutes per day).

**Table 6 nutrients-11-02977-t006:** Association between intake of food groups and low HDL-C in cALL survivors.

Food Groups	Odd Ratio	95% CI	*p* Value
Meat			
Tertile 2 vs. Tertile 1	0.572	0.23–1.40	0.222
Tertile 3 vs. Tertile 1	0.277	0.09–0.83	0.022
Fish and seafood			
Tertile 2 vs. Tertile 1	1.166	0.49–2.80	0.731
Tertile 3 vs. Tertile 1	0.630	0.24–1.63	0.339
Dairy			
Tertile 2 vs. Tertile 1	0.886	0.36–2.18	0.792
Tertile 3 vs. Tertile 1	1.155	0.43–3.09	0.775
Fat			
Tertile 2 vs. Tertile 1	1.179	0.48–2.92	0.722
Tertile 3 vs. Tertile 1	1.581	0.57–4.39	0.379
Vegetables			
Tertile 2 vs. Tertile 1	1.165	0.44–3.07	0.757
Tertile 3 vs. Tertile 1	1.282	0.46–3.54	0.632
Legumes			
Tertile 2 vs. Tertile 1	1.016	0.41–2.51	0.971
Tertile 3 vs. Tertile 1	0.902	0.39–2.08	0.809
Fruits			
Tertile 2 vs. Tertile 1	0.261	0.10–0.70	0.008
Tertile 3 vs. Tertile 1	0.920	0.38–2.24	0.854
Fast food			
Tertile 2 vs. Tertile 1	2.405	1.03–5.63	0.043
Tertile 3 vs. Tertile 1	2.260	0.85–6.03	0.104

HDL-C, high-density lipoprotein cholesterol; cALL: childhood acute lymphoblastic leukemia; OR, odds ratio; CI, confidence interval. Multivariable logistic regression model adjusted for BMI (kg/m^2^), age at diagnosis (years), age at diagnosis squared (years), sex (female), total energy intake (kcal) and moderate-to-vigorous physical activity (minutes per day). Food groups were fitted into a single model except for fast food that was analyzed separately.

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
