# Peer review of "Dietary Intakes Are Associated with HDL-Cholesterol in Survivors of Childhood Acute Lymphoblastic Leukaemia"

_nutrients, 2019, doi:10.3390/nu11122977_

Round 1

Reviewer 1 Report

The article by Morel at al entitled “Dietary intakes are associated with HDL-cholesterol in survivors of childhood acute lymphoblastic leukaemia” examined the associations between food/nutrient intake and the levels of HDL-C in a cohort of cALL survivors. Forty-one % of the participants had dyslipidaemia, most frequently in the form of low HDL-C values (23%). The authors assessed eating habits and physical activity of the subjects and performed multivariate logistic regression analysis to evaluate the associations between particular macro- and micronutrient intake and low HDL-C. High intake of proteins, zinc, copper, selenium, niacin and riboflavin was associated with low HDL-C fraction. The authors state that meat consumption had a positive- and fast food a negative effect on HDL-C. The article is clearly written and easy to follow. However, it is a purely descriptive work, thus all associations are statistical and not empirical as participants were not asked to change their diet for a duration of time to confirm the outcome of the study.

Major points:

I have a problem with the term “protection” as there is no experiment showing me that eating meat, for example, improves HDL-C value. Am I wrong in thinking this? There is an incongruence between one of the statements in the abstract and the results section. In particular, starting from line 29 onwards the abstract reads as follows:

“Low HDL-C was associated with high intake (3rd vs. 1st tertile) of several nutrients: proteins (OR: 0.27, 95% CI: 0.08-0.92), zinc (OR: 0.26, 95% CI: 0.08-0.84), copper (OR: 0.34, 95%CI: 0.12-0.99), selenium (OR: 0.17, 95% CI: 0.05-0.59), niacin (OR: 0.25, 95% CI: 0.08-0.84), riboflavin (OR: 0.31, 95% CI: 0.12-0.76) and vitamin B12 (OR: 32 0.35, 95% CI: 0.13-0.90).”

While lines 187 onwards say:

“The multivariable logistic regression analysis showed that higher intakes in several nutrients (3rd vs. 1st tertile) were protective against low HDL-C presentation (Tables 3, 4 & 5). These were: proteins [odd ratio (OR): 0.29, 95% CI: 0.08-1.00, P=0.05], zinc (OR: 0.26, 95% CI: 0.08-0.84, P<0.05), copper (OR: 0.27, 95% CI: 0.09-0.81, P<0.05), selenium (OR: 0.16, 95% CI: 0.05-0.59, P<0.01), niacin (OR: 0.26, 95% 191 CI: 0.08-0.88, P<0.05), and riboflavin (OR: 0.25, 95% CI: 0.07-0.86, P<0.05).”

In the lines 293-4, the authors talk about small interindividual differences in physical activity. The difference between 8 minutes and 40 minutes is 5 times, I am not sure that this qualifies as a small difference. I agree that the level of physical activity is too low to see an association with HDL-C. In the lines 334-5, the authors say that the fact that less than 60% of participants did consume legumes on the regular basis and suggest lack of association between legume consumption and HDL-C was due to this fact as well as to low consumption of this source of vegetal proteins. I agree with the latter, but imagine that should the consumption be higher than the reported, 60% of participants would be enough to see the association.

Minor points:

Please replace “intake in” with “intake of”. Be more specific in the sentence in line 283 and say that high BMI was associated with low HDL-C. The noun “evidence” (line 312) is a mass noun. There is no such thing as evidences. One would normally talk about “lines of evidence” or “pieces of evidence”.

Reviewer 2 Report

In this manuscript titled “Dietary intakes are associated with HDL-cholesterol in survivors of childhood acute lymphoblastic leukemia”; Morel et al investigated the potential correlation between HDL-cholesterol level and nutrient uptake in the ALL survivors and they report the protective effect of certain macro and micronutrients in terms of controlling the optimum level of HDL-cholesterol. The current study is a pure population-based study with a medium cohort of survivors.  The question that is being addressed in this manuscript is highly important and interesting, however, there are some flaws in the terms of not comparing with a normal population of the same age group without a history of cancer. In general authors have tried to address a very relevant question, however, more information is needed to support their conclusion. Considering the immediate need for increasing our understanding of the impact of nutrients on cancer survivors, it is highly interesting to assess the individual contribution of various micro and macro-nutrients on their health.  Altogether, their preliminary results are interesting however, authors need more evidence to establish their conclusion.

My Specific comments are as follows:

Authors conclude that certain nutrients like protein, meat, selenium, zinc, copper, riboflavin, and niacin have a protective role in HDL-C levels without giving any evidence on the role of these nutrients on the normal individual without ALL history. In such a case, it would be difficult to say that these effects are exclusive to ALL survivors or these effects have nothing to do with their disease background. Since B-ALL is very diverse with respect to translocation and clinical-pathological pattern; it would be needed to provide information on subtypes of ALL in the patients used in this study. This may provide an interesting link between impact nutrients and a particular subtype of ALL. What is the conclusion of figure 1? I am unable to understand the X-axis in the graphs shown. If it does not indicate anything then what is the need for this graph?

Round 2

Reviewer 2 Report

In the revised version of "Dietary intakes are associated with HDL-cholesterol in survivors of childhood acute lymphoblastic leukemia", authors have included certain suggestions. Although, they could not include the normal control without a cancer history, however not at least they clearly mentioned it in the manuscript.